# SARS-CoV-2 antibody seroprevalence and associated risk factors in an urban district in Cameroon

Kene Nwosu [1,10 ✉], Joseph Fokam[2,3,10], Franck Wanda[4], Lucien Mama[5], Erol Orel[1], Nicolas Ray [1,6], Jeanine Meke[4], Armel Tassegning[4], Desire Takou[2], Eric Mimbe [7], Beat Stoll[1], Josselin Guillebert[8], Eric Comte[1,9], Olivia Keiser[1] & Laura Ciaffi [7,9]

The extent of SARS-CoV-2 circulation in many African countries remains unclear, underlining the need for antibody sero-surveys to assess the cumulative attack rate. Here, we present the results of a cross-sectional sero-survey of a random sample of residents of a health district in Yaounde, Cameroon, conducted from October 14 to November 26, 2020. Among the 971 participants, the test-adjusted seroprevalence of anti-SARS-CoV-2 IgG antibodies was 29·2% (95% CI 24·3–34·1). This is about 322 times greater than the 0.09% nationwide attack rate implied by COVID-19 case counts at the time. Men, obese individuals and those living in large households were significantly more likely to be seropositive, and the majority (64·2% [58·7–69·4]) of seropositive individuals reported no symptoms. Despite the high seroprevalence, most of the population had not been infected with SARS-CoV-2, highlighting the importance of continued measures to control viral spread and quick vaccine deployment to protect the vulnerable.

[1] Institute of Global Health, University of Geneva, Geneva, Switzerland. [2] Chantal BIYA International Reference Centre for Research on HIV/AIDS Prevention and Management, Yaounde, Cameroon. [3] Department of Medical Laboratory Sciences, Faculty of Health Sciences, University of Buea, Buea, Cameroon. [4] Centre International de Recherches, d'Enseignements, et de Soins (CIRES), Akonolinga, Cameroon. [5] Health District of Cite Verte, Regional Delegation of Public Health, Yaounde, Cameroon. [6] Institute for Environmental Sciences, University of Geneva, Geneva, Switzerland. [7] Site de Coordination ANRS Cameroun, Hopital Central de Yaounde, Yaounde, Cameroon. [8] Deutsche Gesellschaft für Internationale Zusammenarbeit (GIZ) GmbH, Yaounde, Cameroon. [9] Association de Soutien aux Centres de Recherches, d'Enseignements et de Soins (ASCRES), Geneva, Switzerland. [10] These authors contributed equally: Kene Nwosu, Joseph Fokam. ✉ email: kenechukwu.nwosu@unige.ch

The 2019 coronavirus disease (COVID-19) has placed an unprecedented burden on health systems around the world. In resource-limited settings within sub-Saharan Africa (SSA), gaps in medical infrastructure, difficulties in implementing hygiene measures and perceived public health vulnerabilities were projected to lead to overwhelming morbidity and mortality burdens[1,2].

At the time of writing, however, official counts of COVID-19 cases and deaths have suggested a relatively mild epidemic trajectory on the African continent. As of March 4, 2021, only two African countries, Egypt and South Africa, have reported more than 9000 COVID-19-related deaths[3]. Cameroon, which reported its first case on March 6, 2020, had reported only 35,714 cases 1 year after, implying an attack rate of 1.43 cases per thousand residents (as compared with the 50.7 cases per thousand seen in the European Union).

Multiple hypotheses have been advanced to explain the seemingly mild trajectory of the COVID-19 epidemic in Africa: researchers have pointed to warm climate conditions across much of the continent, timely and effective preventive measures put in place by governments, the young and predominantly rural population, and cross-reactive immunity from other infections as potential mitigating factors[2,4]. However, the true scale of the epidemic in many African countries is still unclear, as the PCR and antigen-confirmed case counts that are commonly relied upon may understate viral spread[2,5].

In this context, the use of serological antibody tests to detect past exposure to the severe acute respiratory syndrome coronavirus 2 (SARS-CoV-2) is valuable. Serological assays can detect evidence of SARS-CoV-2 infection from 2 weeks to several months after the onset of symptoms, and can reveal past infection even in asymptomatic cases[6,7]. They are therefore valuable for accurately assessing the cumulative attack rate—the proportion of the population that has ever been infected with SARS-CoV-2.

Estimates of the SARS-CoV-2 attack rate have important implications for public health policy. They permit a retrospective assessment of the effectiveness of public health control measures; they provide evidence on whether large-scale spread—an additional wave of infection—remains possible; they yield insights into population-specific disease severity; and they inform the strategic deployment of testing, therapies and vaccines. However, only a few SARS-CoV-2 antibody serosurveys have been carried out in African countries to date[8–14], and the majority of serosurveys have been conducted on healthcare workers, convenience samples of blood donors and other non-representative populations.

In this work, we report the results of a cross-sectional, community-based sero-survey of a random sample of residents in a health district of Yaounde, the capital city of Cameroon. We aimed to estimate the prevalence of anti-SARS-CoV-2 antibodies in this population, to assess risk factors for seropositivity, and to investigate the symptoms of seropositive respondents.

## Results

**Characteristics of the enrolled sample.** Out of the 255 households visited between October 14 and November 26, 2020, 180 (70.6%) agreed to participate, resulting in a final sample of 971 participants (full study profile in Supplementary Figs. 1 and 2). Table 1 shows the sociodemographic characteristics of the final sample. The median age of participants was 26 years (IQR: 14–38), and 56.5% of them were female ($n = 549$). The majority were students (39.3%, $n = 402$), informal workers (21.3%, $n = 218$) or traders (12.6%, $n = 129$). A total of 112 respondents (11.5%) reported suffering from a chronic condition, mainly

| Table 1 Sample characteristics. | | |
|---|---|---|
| **Characteristic** | **N** | **%** |
| Age groups (years) | | |
| 5–14 | 241 | 24.8 |
| 15–29 | 325 | 33.5 |
| 30–44 | 212 | 21.8 |
| 45–64 | 153 | 15.8 |
| 65+ | 40 | 4.1 |
| Sex | | |
| Female | 549 | 56.5 |
| Male | 422 | 43.5 |
| BMI (kg/m²) | | |
| <18.5 (Underweight) | 160 | 16.5 |
| 18.5–24.9 | 400 | 41.2 |
| 25–30 (Overweight) | 247 | 25.4 |
| >30 (Obese) | 160 | 16.5 |
| Unknown | 4 | 0.4 |
| Education level | | |
| Secondary | 433 | 44.6 |
| Primary | 318 | 32.7 |
| University | 145 | 14.9 |
| No formal instruction | 52 | 5.4 |
| Doctorate | 17 | 1.8 |
| Other | 6 | 0.6 |
| Profession | | |
| Student | 402 | 39.3 |
| Informal worker | 218 | 21.3 |
| Trader | 129 | 12.6 |
| Home-maker | 74 | 7.2 |
| Unemployed | 70 | 6.8 |
| Salaried worker | 54 | 5.3 |
| Retired | 32 | 3.1 |
| Other | 43 | 4.2 |
| Chronic conditions | | |
| Hypertension | 32 | 3.3 |
| Respiratory illness | 17 | 1.7 |
| Diabetes | 11 | 1.1 |
| Other | 52 | 5.3 |

Sociodemographic characteristics of the participants in the final sample of 971 study participants. N is the number of individuals in each stratum.
*BMI* body mass index.

hypertension (3.3%, $n = 32$), respiratory illnesses (1.7%, $n = 17$) or diabetes (1.1%, $n = 11$).

**Crude seroprevalence.** Figure 1 shows the unadjusted seroprevalence of anti-SARS-CoV-2 IgG and IgM antibodies in the study sample. Of the 971 respondents tested for antibodies, 302 (31.1%) were IgG positive, 32 (3.3%) were IgM positive and a combined 328 (35.1%) were positive for at least one antibody type (Fig. 1a). The overlap between IgG and IgM seropositivity was low, with only six individuals testing positive for both antibody types. Seropositivity estimates did not change significantly from week to week during the six weeks of survey completion (Supplementary Fig. 3). Despite the high seroprevalence of antibodies, active COVID-19 infection was uncommon: only one PCR test was positive among the 21 tests performed on suspected cases, for an implied active infection rate of 0.1%.

At the household level, the range of seroprevalence was broad: from 0 to 100%, with a median of 33% (IQR: 0–50%) (Fig. 1c). Most households (73%, 131 of 180) had at least one IgG and IgM seropositive resident. Notably, only in two households (1.1%) was everyone seropositive; one of these was a single-resident household and the other had two residents. The detailed distribution of household seropositivity is reported in Supplementary Fig. 5.

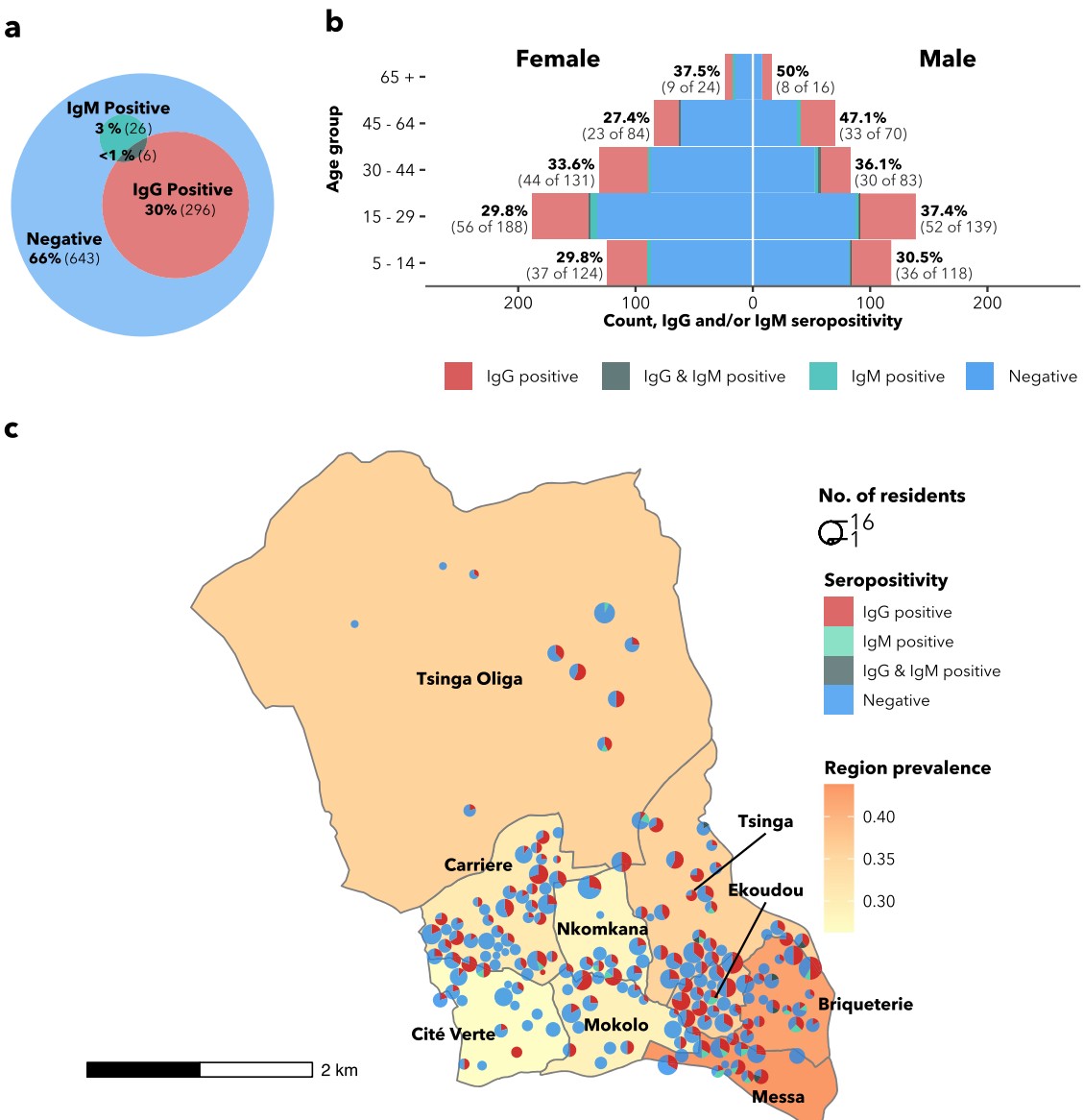

**Fig. 1 Crude IgG and IgM seroprevalence. a** Euler diagram showing seropositivity of respondents by antibody test. **b** Seropositivity of respondents by antibody type and age-sex stratum. Percentage labels and numbers in parentheses indicate the proportion of each stratum that is IgG and/or IgM seropositive. **c** Household and geographic variation in seropositivity. Choropleth fill colour indicates the neighbourhood seroprevalence (IgG and/or IgM). Pie charts indicate household size, household location and the proportion of the household that is seropositive. Pie charts are dodged and jittered to avoid overlap and to preserve location anonymity. Five households are not shown due to improperly-coded or missing coordinates. IgG Immunoglobulin G, IgM Immunoglobulin M.

**Adjusted seroprevalence.** Because men were slightly under-sampled in the survey and seroprevalence was higher for men, the application of age-sex sample weights raised the overall seroprevalence estimates marginally, from 31.1 to 31.3% (95% CI 27.9–34.9; Table 2 and Supplementary Fig. 4). The adjustments for sensitivity and specificity of the antibody test then moved the final estimates downwards, for a final adjusted IgG seroprevalence of 29.2% (95% CI 24.3–34.1; Table 2). Men had a higher seroprevalence than women (33.1% [27.6–40.5] versus 25.3% [20.0–31.2]), and seroprevalence increased with age, but these sex and age differences were not statistically significant.

The proportion of IgM-positive individuals was lower (3.3%) than the expected false positive rate of the IgM test (6.9%), so adjusted IgM seroprevalence estimates were statistically indistinguishable from zero. For this reason, adjusted IgM seroprevalence is not reported, and the IgM results were not considered in the analysis of symptoms or of seropositivity risk factors.

**Risk factors for seropositivity.** As shown in Fig. 2, the multivariable risk factor analysis for IgG seropositivity revealed significantly higher odds of seropositivity for men (OR: 1.6 [95% CI 1.2–2.2]), residents of households with six or more residents (OR: 1.6 [1.1–2.4]; reference: households with three to five residents) and individuals with a BMI above 30 kg/m² (OR: 1.8 [1.1–2.8]; reference: 18.5–24.9 kg/m²). The highest stratified seroprevalence was seen in respondents who had been in contact with a known or suspected COVID-19 case: 45.7% (16 of 35) of these individuals were IgG positive, but the odds ratio was not significant.

**Table 2 Age-sex weighted and test-adjusted seroprevalence estimates for anti-SARS-CoV-2 IgG antibodies.**

| | *n* | Seropos. | Seroprevalence (95% confidence interval) | | |
| --- | --- | --- | --- | --- | --- |
| | | | Crude | Age-sex-weighted | Age-sex-weighted, test-adjusted |
| Overall | 971 | 302 | 31.1% (27.8–34.6) | 31.3% (27.9–34.9) | 29.2% (23.8–34.9) |
| Sex | | | | | |
| Female | 549 | 154 | 28.1% (23.7–32.8) | 28.0% (23.6–32.9) | 25.3% (19.0–32.2) |
| Male | 422 | 148 | 35.1% (30.3–40.1) | 34.6% (29.9–39.7) | 33.1% (26.4–40.5) |
| Age group | | | | | |
| 5–14 | 241 | 69 | 28.6% (22.2–36.0) | 28.7% (22.3–36.0) | 26.1% (17.6–35.6) |
| 15–29 | 325 | 98 | 30.2% (25.3–35.5) | 30.7% (25.8–36.0) | 28.5% (21.7–35.9) |
| 30–44 | 212 | 69 | 32.5% (26.4–39.4) | 32.7% (26.2–39.9) | 30.8% (22.3–40.4) |
| 45–64 | 153 | 51 | 33.3% (26.4–41.1) | 34.1% (27.1–41.8) | 32.5% (23.4–42.6) |
| 65+ | 40 | 15 | 37.5% (23.1–54.6) | 39.4% (24.4–56.6) | 38.7% (20.6–59.9) |
| Neighborhood | | | | | |
| Cité Verte | 72 | 16 | 22.2% (13.2–34.9) | 22.1% (13.1–34.8) | 18.4% (6.9–33.8) |
| Briqueterie | 106 | 37 | 34.9% (25.9–45.2) | 33.7% (25.2–43.3) | 32.0% (21.4–44.2) |
| Carriere | 236 | 72 | 30.5% (23.6–38.4) | 31.1% (24.0–39.3) | 29.0% (19.7–39.5) |
| Ekoudou | 190 | 65 | 34.2% (26.7–42.6) | 34.8% (27.4–43.0) | 33.3% (24.0–44.0) |
| Messa | 48 | 17 | 35.4% (18.0–57.9) | 35.7% (17.8–58.8) | 34.4% (12.9–62.4) |
| Mokolo | 96 | 27 | 28.1% (18.1–40.9) | 28.8% (17.7–43.4) | 26.3% (12.6–44.1) |
| Nkomkana | 75 | 18 | 24.0% (13.7–38.6) | 22.9% (12.6–37.9) | 19.3% (6.6–37.5) |
| Tsinga | 81 | 28 | 34.6% (26.8–43.3) | 35.3% (27.0–44.5) | 33.9% (23.5–45.7) |
| Tsinga Oliga | 67 | 22 | 32.8% (17.3–53.2) | 32.5% (16.7–53.6) | 30.6% (11.6–56.1) |

Confidence intervals for the crude- and age-sex-reweighted estimates are Wald-type intervals, while confidence intervals for the test-adjusted estimates were calculated by bootstrap sampling (see Methods).

| | n | Pos. | % Pos. | Univariate OR (95% CI) | Multivariate OR (95% CI) | Multivariate OR plot |
| --- | --- | --- | --- | --- | --- | --- |
| **Sex** | | | | | | |
| Female | 545 | 153 | 28.1 | Reference | Reference | |
| Male | 421 | 148 | 35.2 | 1.4 (1.1 - 1.9) | 1.62 (1.19 - 2.2) | * |
| **Age** | | | | | | |
| 5 - 14 | 239 | 69 | 28.9 | 0.87 (0.57 - 1.3) | 0.98 (0.56 - 1.7) | |
| 15 - 29 | 324 | 98 | 30.2 | 0.89 (0.6 - 1.3) | 1.06 (0.7 - 1.6) | |
| 30 - 44 | 211 | 68 | 32.2 | Reference | Reference | |
| 45 - 64 | 152 | 51 | 33.6 | 1.05 (0.66 - 1.7) | 0.96 (0.59 - 1.6) | |
| 65 + | 40 | 15 | 37.5 | 1.33 (0.63 - 2.8) | 1.25 (0.59 - 2.7) | |
| **BMI group** | | | | | | |
| < 18.5 (Underweight) | 160 | 45 | 28.1 | 0.97 (0.63 - 1.5) | 0.9 (0.52 - 1.5) | |
| 18.5 - 24.9 | 400 | 115 | 28.7 | Reference | Reference | |
| 25 - 30 (Overweight) | 246 | 81 | 32.9 | 1.2 (0.84 - 1.7) | 1.23 (0.83 - 1.8) | |
| > 30 (Obese) | 160 | 60 | 37.5 | 1.53 (1.02 - 2.3) | 1.75 (1.09 - 2.8) | * |
| **Contact with international traveler** | | | | | | |
| No contact with traveler | 803 | 245 | 30.5 | Reference | Reference | |
| Recent contact with traveler | 103 | 30 | 29.1 | 0.91 (0.56 - 1.5) | 0.78 (0.47 - 1.3) | |
| Unsure about traveler contact | 60 | 26 | 43.3 | 1.82 (1.02 - 3.2) | 1.69 (0.92 - 3.1) | |
| **Contact with COVID case** | | | | | | |
| No COVID contact | 701 | 202 | 28.8 | Reference | Reference | |
| Recent COVID contact | 35 | 16 | 45.7 | 2.2 (1.1 - 4.6) | 2.13 (0.99 - 4.6) | |
| Unsure about COVID contact | 230 | 83 | 36.1 | 1.4 (1 - 2) | 1.38 (0.96 - 2) | |
| **Number of household members** | | | | | | |
| 1 - 2 | 20 | 2 | 10 | 0.29 (0.06 - 1.4) | 0.3 (0.06 - 1.4) | |
| 3 - 5 | 238 | 64 | 26.9 | Reference | Reference | |
| > 5 | 708 | 235 | 33.2 | 1.39 (0.96 - 2) | 1.58 (1.07 - 2.3) | * |

**Fig. 2 Risk factor analysis for crude IgG seropositivity n = 966.** Based on logistic models with household random intercepts. Box-whisker plot indicates odds ratio and 95% confidence interval. Asterisks indicate significance at a 0.05 alpha level. Five individuals were dropped due to missing covariables. Recent contact indicates contact since March 1, 2020. Variables that were found not to be significant at a 0.30 alpha level, and which were not controlled for in the multivariable regression, include: presence of comorbidities, breadwinner status, adherence to social distancing rules, household neighbourhood and presence of children in the household. BMI body mass index, n number of individuals, Pos positive, OR odds ratio, CI confidence interval.

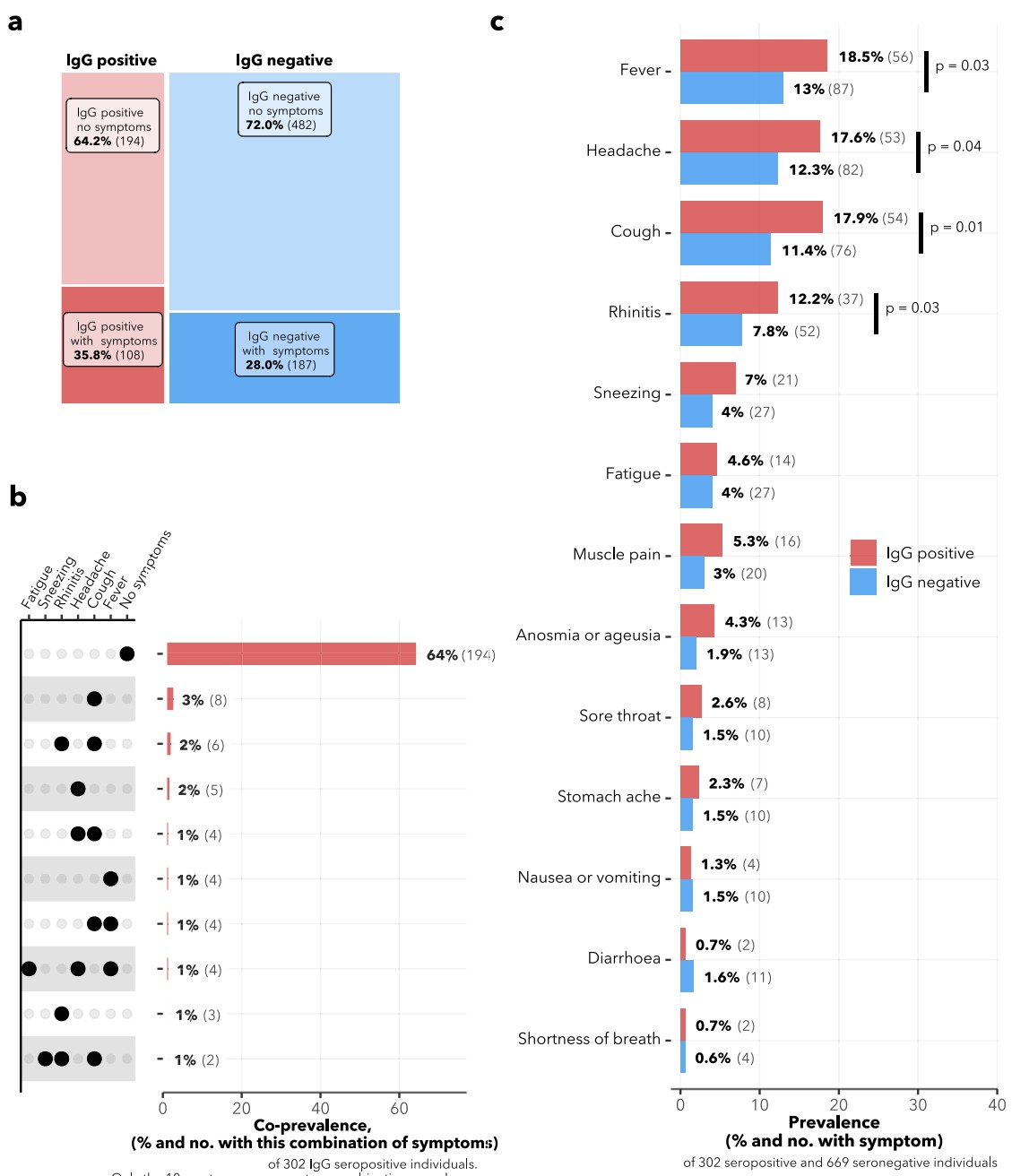

**Fig. 3 COVID-compatible symptoms of survey participants.** Participants reported any COVID-compatible acute symptoms (all shown in panel **c**), which were experienced between March 1, 2020 and the date of survey. **a** Matrix plot showing the intersection of symptomaticity with IgG seropositivity. The area of each rectangle is proportional to the number of respondents in the category. **b** The ten most common symptom profiles among IgG seropositive individuals. **c** Comparison in frequency of symptoms between IgG seropositive and seronegative individuals. $\chi$-square p-values are shown for significant differences between seropositive and seronegative symptomatic proportions.

**Symptoms and health-seeking behaviour.** Among the 302 IgG seropositive participants, 35.8% ($n = 108$) reported having had at least one COVID-19-related symptom; among the 669 IgG seronegative participants, this proportion was 28.0% ($n = 187$) (Fig. 3a). The most common symptoms reported among the IgG seropositive individuals were fever (18.5%, $n = 56$), headache (17.6%, $n = 53$), cough (17.9%, $n = 54$) and rhinorrhea (12.3%, $n = 37$), and all four were significantly more common in seropositive than in seronegative individuals (Fig. 3c). Surprisingly, anosmia and/or ageusia was only experienced by 4.3% ($n = 13$) of the seropositive respondents. Cough alone and cough plus rhinorrhea were the two most common symptom profiles among

IgG seropositive participants (Fig. 3b). In terms of severity, 80% of IgG seropositive respondents with symptoms (83 of 104) graded these symptoms as mild or moderate.

Among the 302 IgG seropositive individuals, only 27 (8.9%) consulted any healthcare services over the pandemic period (Supplementary Fig. 6). The most common medications taken by the IgG seropositive respondents were paracetamol (19.9%, $n = 60$), traditional medicines (14.6%, $n = 44$) and antibiotics (10.3%, $n = 31$; Fig. 4), and these were most commonly self or family-prescribed.

A total of 46 respondents reported having been hospitalized between March 1, 2020 and the date of survey, but only one of

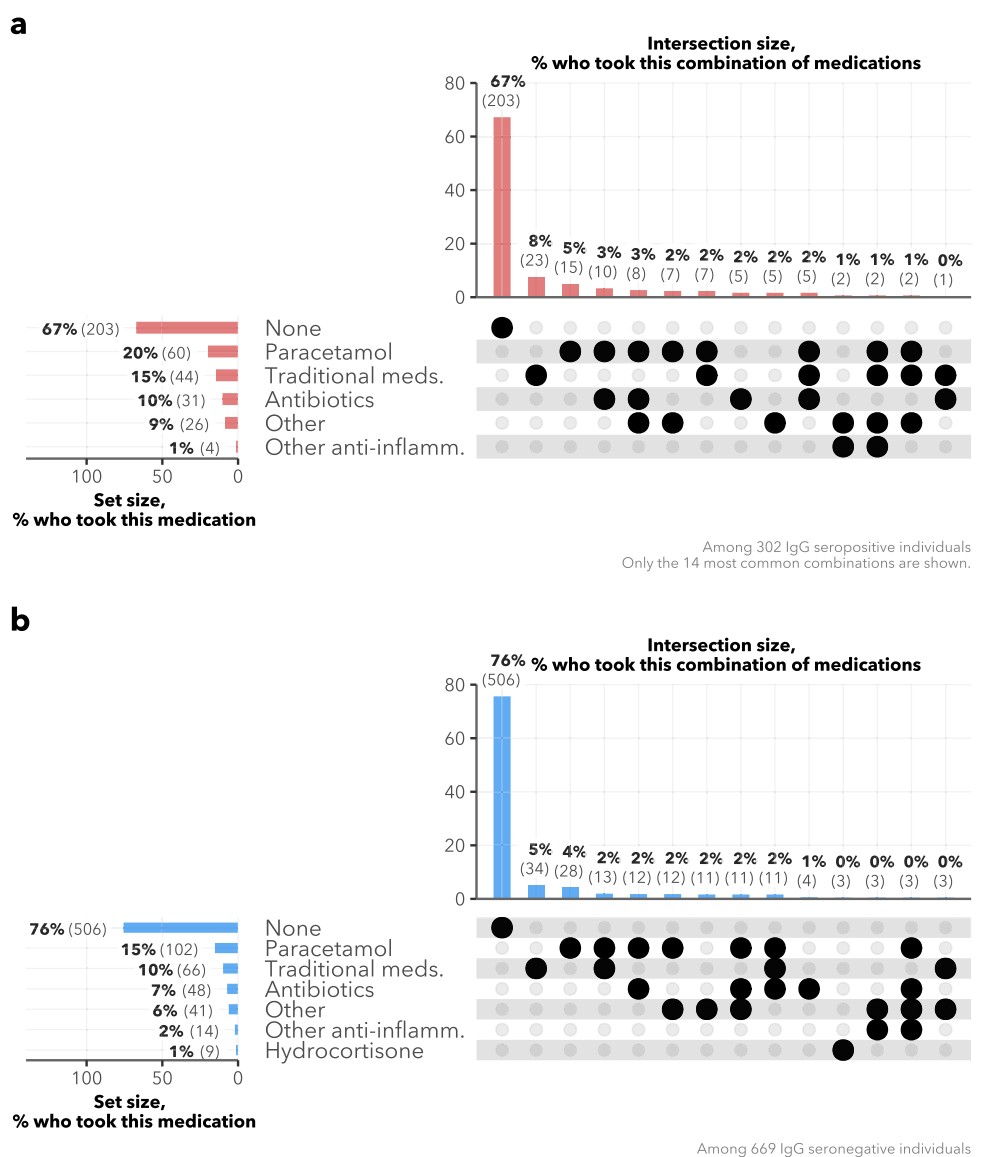

**Fig. 4 Medications taken.** Upset plots showing medications taken by **a** IgG seropositive and **b** IgG seronegative individuals, between March 1, 2020 and the date of survey. Horizontal bars to the left represent the percentage of respondents in each group who took the given medication. Vertical bars show the percentage of respondents in each group who took the combination of medications marked with the black dot(s) in that column. Meds: Medicines. Other anti-inflamm other anti-inflammatory drugs.

these was reported to be COVID-19 related, implying a hospitalization rate of 0.3% (one out of 302 IgG seropositive respondents). Over the same period, 11 of the 180 surveyed households reported the death of a family member, but none of these deaths was reported to be COVID-19-linked.

## Discussion

In this urban setting of Cameroon, the adjusted seroprevalence of SARS-CoV-2 IgG antibodies was found to be 29.2%, implying that around 126,000 of the district's 432,858 inhabitants had been infected with SARS-CoV-2 by the survey's end date, November 26, 2020. It is notable that by this date, only 24,002 cases had been officially reported in the entire country, which has a population of 26.55 million people[15]. Thus, the measured seroprevalence in the surveyed district was about 322 times greater than the nationwide attack rate implied by PCR- and antigen-confirmed case counts at the time[3]. While this

large discrepancy may indicate a particularly high attack rate in Yaounde, it also likely points to inadequate testing, and suggests that the true cumulative incidence of COVID-19 in Cameroon may be much higher than the number of cases officially reported.

The underreporting of COVID-19 cases implied by our survey is not unique. In a recent systematic review, Chen et al. (2021) compared the number of infections estimated by seroprevalence surveys to the number of PCR-confirmed infections in a range of countries and found a pooled ratio of 11.1 (95% CI 8.3–14.9)[16], meaning that for each virologically confirmed COVID-19 case, there were at least ten undetected infections in the community. Across individual settings, this ratio varied widely, from 2.0 in a Faroe Islands study[17], to 103.0 in a study of Indian villages[18]. Taken together, these findings and ours suggest that PCR-confirmed case counts are poor proxies for the true attack rate of SARS-CoV-2, and that cross-national comparisons based on such case counts may be misleading.

We found that men and obese individuals (BMI > 30 kg/m²) were significantly more likely to be seropositive, and we also observed higher seropositivity, although non-significant, among older age groups. It is uncertain whether the raised seroprevalence in these groups represents a greater risk of SARS-CoV-2 infection per se, or a greater probability of antibody detection. Older, male and obese individuals are known to experience more severe COVID-19 symptoms[19], and severe illness is linked to stronger and longer-lasting antibody responses[20]. As a result, serosurveys performed several months after infection may detect antibodies more frequently in these groups because they experienced more severe illness and stronger antibody responses, not because they were infected at higher rates.

Alternatively, the physiological factors that predispose men, obese individuals and older people to more severe disease may also make them more susceptible to initial infection. Some studies have suggested that adults may be more likely to be infected with SARS-CoV-2 than young children[21,22], and a few point prevalence studies have found slightly raised viral attack rates in men[23,24]. If the risk factors for infection and those for severe illness overlap, then surveillance and prevention measures that focus on the higher-risk groups may be particularly appropriate, especially in contexts where stringent population-wide measures are not feasible.

The rate of asymptomatic infection in our study is higher than usually described; ~70% of the IgG positive individuals in the sample did not report any COVID-19-related symptoms. In a recent meta-analysis by Byambasuren et al.[25], the measured asymptomatic rate was much lower—a pooled estimate of 17% (95% CI 14–20%). COVID-19-related hospitalization was also relatively uncommon in our sample (0.3% among the IgG seropositive individuals), and no COVID-19-linked deaths were reported in any of the surveyed households.

These favourable outcomes could reflect the relatively young population in the region of study. As COVID-19 severity increases with age, the overall burden of disease in young populations is expected to be less severe[19]. Cameroon's median age of 18.6 years and the African median of 19.7[26] are therefore noteworthy, and may explain the limited COVID-19 mortality impact here as compared with the other regions; the median age in Europe, for example, is 40.2 years[26].

However, caution should be exercised in interpreting the low hospitalization and death rates implied by our study. The surveyed households reported a total of 46 hospitalizations and 11 family member deaths over the pandemic period. While only one hospitalization and none of the deaths were known to be COVID-19 related, it is possible that the factors limiting testing in the general population also applied to those who were hospitalized and dying. Thus, we cannot rule out the possibility that some of these hospitalizations or deaths were actually COVID-19-linked. Of note, a study of deceased patients in a hospital morgue in Lusaka, Zambia found that 15% of those who died between June and September 2020 had COVID-19 at the time of death, and very few of them were tested for SARS-CoV-2 before death[27]. Further investigations are therefore required to assess the number of undiagnosed COVID-19-related deaths in countries within the SSA region.

Our study has several major strengths. This is one of the first studies to assess SARS-CoV-2 antibody seroprevalence in a random sample of residents in an African city. Our random selection procedure ensures representativeness of the target population and minimizes the risk of bias. The study also demonstrates the feasibility of performing a geo-sampled door-to-door serological survey in an African city—a simple, effective study design that can be applied widely. Finally, we validated the performance of the chosen antibody test on local pre-pandemic sera, thus ruling out concerns about low test specificity in African populations[28].

The study was also subject to a number of limitations. We registered a household refusal rate of 24%, which may be a source of bias if household refusal was correlated with seropositivity. Second, we asked participants to recall symptoms experienced over a period of seven to eight months, which may have led to recall bias. This long time interval also meant that we were unable to directly link reported symptoms to COVID-19 infection: many of the reported symptoms may have been caused by other illnesses experienced over the same time period.

A further limitation concerns the sensitivity value assumed for the IgG antibody test. This value was obtained from a validation study conducted with hospitalized COVID-19 patients, who are likely to have experienced more severe illness and greater antibody responses than our study population[29]. In addition, that study was conducted on samples obtained within two months of symptom onset, a short time frame over which minimal antibody waning would be expected. Our study, in contrast, was conducted about four months after the first epidemic peak in Cameroon, at which point the antibody levels of those infected may have dropped[30]. These two dissimilarities suggest that the assumed sensitivity may be an overestimate in the context of our study population. If that is the case, our final adjusted estimate may be downwardly biased.

In conclusion, our sero-survey indicates that around one in three individuals in Yaounde, Cameroon had been infected with SARS-CoV-2 by November 26, 2020. Together with similarly high seroprevalence estimates from other SSA studies—24.5% in Niger state, Nigeria[8], 25.1% in Abidjan, Ivory Coast[14], 19.7% in Brazzaville, Congo[31], among others—this finding points to extensive and under-reported circulation of SARS-CoV-2 in settings across the African continent. As men, obese individuals, and those living in large households were found to be significantly more affected, it may be valuable to tailor public health interventions toward these groups. Despite the high seroprevalence, the data indicate that in Yaounde, as in most other surveyed regions in Africa, the majority of the population has avoided SARS-CoV-2 infection at the time of writing, highlighting the importance of continued mitigation measures, tracing and testing and quick vaccine deployment to curb further spread.

## Methods

**Population and sampling.** The study was conducted in Cité Verte, a health district of Yaounde, Cameroon with an estimated population of 432,858 inhabitants.

We used a single-stage cluster sampling design with a target population of 250 households, and planned to interview all residents of each household—an estimated 1000 individuals. Households were randomly selected from a pre-processed set of residential buildings based on OpenStreetMap data (full procedure in appendix 1 p 7)[32]. Data collection took place between October 14 and November 26, 2020 (sampling timeline in appendix 1 p 2). In the field, each sampled household was visited by study investigators, who either interviewed residents on the first meeting, or arranged an appointment for a future interview if household members were not all present.

In each household, all individuals between five and 80 years of age were included if they (a) had been present in the household for at least 14 days prior to the survey, and (b) could give written informed consent (or had an adult guardian who could give consent).

**Testing procedure.** The Abbott Panbio™ COVID-19 IgG/IgM Rapid Test Device was used to screen for SARS-CoV-2 IgG and IgM antibodies in capillary blood collected from a finger prick. This is an immunochromatographic, lateral flow test for the qualitative detection of IgG and IgM antibodies to the nucleocapsid (N) protein of SARS-CoV-2. Test results were classified into one of five categories: negative, IgG positive alone (indicating past infection), IgM positive alone (indicating recent infection), IgG and IgM positive (also indicating recent infection) or invalid/inconclusive. Invalid/inconclusive results were repeated and classified accordingly.

The test has a manufacturer-estimated sensitivity and specificity of 95.8% and 94%, respectively. However, since test specificity varies across populations, externally assessed specificity values may be misleading. Thus, we also validated the test specificity on a panel of 246 pre-pandemic (2017) samples from individuals living in Yaounde. The IgG test correctly diagnosed 230 of these negative samples

(93.5% specificity [95% CI: 89.7–96.2]), while the IgM test correctly diagnosed 229 samples (93.1% specificity [95% CI: 89.2–95.9]). For IgG sensitivity, an estimate of 91.5% (83.2–96.5) was used, as obtained from a validation study on hospitalized COVID-19 patients 14–56 days post symptom onset[29].

Alongside serological testing, a questionnaire was administered on disease symptoms experienced since March 1, 2020, and on health-seeking behaviour over the same pandemic period.

**Data analysis.** To arrive at final seroprevalence estimates, crude proportions were first reweighted to match the age-sex distribution of the Yaounde population, as sourced from the 2018 Cameroon DHS[33]. Briefly, the sample was grouped into strata based on age (categorized as 5–14, 15–29, 30–44, 45–64 or 65+ years) and sex (categorized as male or female). Then weights for each age-sex stratum (e.g. males aged 5–14), were obtained by dividing the stratum's actual population proportion by the stratum's proportion in our sample (see Supplementary Table 1). These weights were then used in the calculation of the weighted seroprevalence. Thus, the age-sex-standardized prevalences represent the expected prevalence if the age and sex distribution of our sample mirrored that of the reference population.

We then used the Rogan-Gladen formula to adjust IgG seroprevalence estimates to account for test sensitivity and specificity[34]. We did not apply test performance corrections to the IgM seroprevalence estimates due to the inherently uncertain sensitivity of IgM tests; as IgM antibodies decline rapidly after infection, sensitivity varies widely with time since infection.

Confidence intervals around the crude- and age-sex-reweighted estimated are Wald-type intervals computed on the log-odds scale, as implemented in the r "survey" package[35]; these intervals take the survey's single-stage cluster-sampled design into account. Confidence intervals for the test-adjusted estimates were calculated by bootstrap sampling using the "adjPrevSensSpecCI" function of the "bootComb" R package[36], taking 100,000 parametric bootstrap samples for each estimate. The bootstrapping procedure propagates the uncertainty from the sensitivity and specificity validation studies, as well as the cluster-robust crude seroprevalence uncertainty into the final confidence interval for the adjusted seroprevalence.

For the seropositivity risk factor analysis, we used logistic regression models with household random intercepts to account for within-household clustering. The following risk factors were analysed: sex, age (categorized as 5–14, 15–29, 30–44, 45–64 or 65+ years), highest education level (no formal instruction, primary, secondary, university, doctorate), BMI group (<18.5, 18.5–24.9, 25–30 or >30 kg/m$^2$), contact with an international traveller since March 1, 2020 (recent contact, no contact or unsure about contact), contact with a suspected or confirmed COVID-19 case since March 1, 2020 (recent contact, no contact or unsure about contact), presence of comorbidities (combining hypertension, respiratory illness, diabetes, tuberculosis, HIV, cardiovascular illness and/or "other illnesses", which were not explicitly listed in the questionnaire), whether or not the respondent was the breadwinner, adherence to social distancing rules ("Yes", "No" or "Partly"), location of the household (one of nine neighbourhoods), number of household members, and whether or not there were children in the household. Each variable was first analysed in a univariate model. A Wald Chi-square test was then carried out on each univariate model, and all variables below a relaxed p-value cut-off of 0.30 were entered into the multivariable analysis (p values were not corrected for multiple-hypothesis testing). This full multivariate model was presented. Individuals with missing covariables were not included in the regression analysis.

**Software.** Data were collected on the field with KoboCollect (version 1.29.3-1).

Microsoft BING maps and QGIS 3.16 were used for map creation and processing. A vector-based OpenStreetMap (OSM) data set for Yaounde was obtained from GeoFabrik (http://download.geofabrik.de/africa.html) on 11 September 2020.

Data processing, analysis and visualization was performed with R version 4.0.2. The following R packages were employed in the analysis: bootComb (1.0.1), cAIC4 (0.9), car (3.0–10), carData (3.0–4), coda (0.19–4), cowplot (1.1.1), DescTools (0.99.40), devtools (2.3.2), dplyr (1.0.5), epiR (2.0.19), eulerr (6.1.0), forcats (0.5.1), ggallin (0.1.1), ggforestplot (0.1.0), ggnewscale (0.4.5), ggplot2 (3.3.3), ggspatial (1.1.5), ggtext (0.1.1), glmglrt (0.2.2), gridExtra (2.3), gridGraphics (0.5–1), gt (0.3.0), here (1.0.1), huxtable (5.2.0), inspectdf (0.0.10), ISOweek (0.6–2), janitor (2.1.0), lattice (0.20–41), lme4 (1.1–26), lmerTest (3.1–3), lubridate (1.7.9.2), magrittr (2.0.1), Matrix (1.3–2), nlme (3.1–152), packcircles (0.3.4), pacman (0.5.1), paletteer (1.3.0), patch (0.0.1), patchwork (1.1.1), permute (0.9–5), prevalence (0.4.0), purrr (0.3.4), readr (1.4.0), readxl (1.3.1), renv (0.13.0), reshape2 (1.4.4), rjags (4–10), scales (1.1.1), scatterpie (0.1.5), sf (0.9–7), srvyr (1.0.1), stringi (1.5.3), stringr (1.4.0), styler (1.3.2), survey (4.0), survival (3.2–10), sysfonts (0.8.3), tibble (3.1.1), tidyr (1.1.3), tidyverse (1.3.0), usethis (2.0.1), vegan (2.5–7), viridis (0.5.1), viridisLite (0.3.0) and xlsx (0.6.5).

**Ethical considerations.** The study protocol obtained the ethical clearance (N° 2020/09/1292/CE/CNERSH/SP) and the administrative authorization of the Ministry of Health of Cameroon (N°D30-845/L/MINSANTE/SG/DROS). Every adult participant (21 years or above) signed an informed consent form and, for minors, a person with parental authority was asked to sign the consent form. Minors who were able to sign were also asked to sign a special assent form. In cases where active COVID-19 was suspected (based on the result of the IgG antibody test and self-

reported symptoms), a nasopharyngeal swab test was offered to the respondent and sent for analysis at the study reference laboratory, the Chantal BIYA International Reference Centre (CIRCB) in Yaounde. All members of the survey team were trained in health research ethics and good clinical practice.

**Role of the funding source.** The sponsors of the study had no role in study design, data collection, data analysis, data interpretation or writing of the report.

**Reporting summary.** Further information on research design is available in the Nature Research Reporting Summary linked to this article.

## Data availability
All raw data and map files needed to reproduce the analytic outputs (figures, tables and statistics) in the paper have been archived on Zenodo with the following url: https://doi.org/10.5281/zenodo.5218965. The files are also available at the following GitHub page: https://github.com/kendavidn/yaounde_serocovpop_shared/tree/v1.0.0. Source data are provided with this paper.

## Code availability
All code needed to reproduce the analytic outputs in the paper have also been archived on Zenodo and GitHub within the same repositories as the shared data.

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

## Acknowledgements

This study was supported by the UHC Program and Bilateral Health Project in Cameroon of the Deutsche Gesellschaft für Internationale Zusammenarbeit (GIZ) GmbH, the P4H Health Financing Network, and the Canton of Geneva. O.K. and E.O. were also supported by the Swiss National Science Foundation (grant numbers 163878 and 196270). We would like to thank all the participants who accepted the study staff in their homes and generously offered their time during a difficult period. We thank all the interviewers, the Cameroonian Ministry of Health, the Cameroonian National Ethics Committee and all the staff of the health district of Cité Verte. We are also grateful to: Miss Thomas Eyinga, Evelyne Boyomo and the community health staff for field support; Wingston Ng'ambi and other members of the UNIGE Institute of Global Health for helpful feedback and suggestions; the virology laboratory at CIRCB, especially the Director, Prof. Alexis Njolo, for laboratory support; and Dr Hermine Abessolo, Dr Ngo Nsonga Marie Thèrese and Mr Talom Calice of the Scientific Committee for their valuable guidance.

## Author contributions

L.C., F.W. and E.C. conceived and designed the study. F.W., L.M., A.T. and J.M. participated in data collection. N.R. designed the spatial sampling methodology. J.F. and D.T. validated the testing protocol. O.K., B.S. and J.G. oversaw the study design and execution. K.N. and E.O. analysed and interpreted the data and K.N. produced the output figures and tables. K.N. and J.F. wrote the initial manuscript, and all authors contributed to subsequent revisions and approved the final version submitted for publication. L.C., E.M. and F.W. had full access to all the data in the study, and K.N. and L.C. had final responsibility for the decision to submit for publication.

## Competing interests

The authors declare no competing interests.
