## [Peer Review File · Nature Communications]

SARS-CoV-2 antibody seroprevalence and associated risk factors in an urban district of CameroonREVIEWER COMMENTS

Reviewer #1 (Remarks to the Author):

Nwosu et al. present results of a serological survey in one health province of urban Cameroon. The paper is well written and considered; I agree with the points made in the discussion. It is important research as relatively few serological surveys have been published from the continent of Africa, and none from Cameroon.

The statistical analyses are appropriate, though there are limitations that could be noted.

1) First, since sampling was within households, the survey population are not all independent. This is taken into account in the risk factor analysis, but not in the overall seroprevalence estimates, and could be noted as a potential source of bias.

2) Uncertainty in the sensitivity and specificity are not taken into account. On page 15, lines 272-273, I would like to see confidence intervals given for the sensitivity and specificity used.

3) The hospitalised patients sensitivity is likely to be an over-estimate, firstly because patients will have experienced more severe hospitalised infection - giving rise to greater antibody responses on average, and secondly because all measurements were taken within 2 months - an optimal window during which we would expect no or little waning. The potential effect on adjusted estimates should at least be discussed.

(Note again, I do believe the methods used are appropriate. To amend methods to address the above would be considerably more complex, and would not gain any further understanding.)

The only other minor comment I have was on page 3, lines 70-72. I saw that results of a serological survey on a random sample of the Zambian population was published very recently in *The Lancet Global Health* (Mulenga et al). Please re-check the literature and update this sentence accordingly.

Reviewer #2 (Remarks to the Author):

The manuscript reports the results of a cross-sectional sero-survey in Yaoundé, Cameroon. The authors adjust seroprevalence for test sensitivity and specificity and investigate the risk factors associated with IgG presence. This is one of the few Covid-19 sero-surveys from Africa.

The manuscript is lovely. Very clearly and well written, good exploration of risk factors and results are well discussed. It is great to see such detailed study from Africa, with some interesting results, e.g. very high overall seroprevalence compared to other parts of the world, especially at the time of the study.

My only main concern is the use of both univariate and multivariate analysis on the same dataset. I do not understand why this is done. Implementing the multivariate analysis from the start to all variables available would have been better and avoid multiple testing errors and removing important variables. Unless there was an unexplained reason for this, I would suggest removing the univariate analysis from the manuscript and present only the multivariate with all variables. If required, a backward selection method could be implemented to reduce the number of variables to the most essential ones.

Other suggestions:

- It is not always clear when the authors are talking about raw vs adjusted seroprevalence, especially in the results. I would suggest keeping the raw values in the table as is but focus the text on the adjusted values only. Please make this clear throughout the text.
- Also not clear what the age-sex weighted seroprevalence is, how it is estimated and how it is used.

The methods on this are also not clear.

- Is the logistic regression implemented on the adjusted seroprevalence?

Reviewer #3 (Remarks to the Author):

Thank you for the opportunity to review the paper titled "SARS-CoV-2 antibody seroprevalence and associated risk factors in an urban district of Cameroon." This work is incredibly important, as the authors note that seroprevalence studies from Africa are rare. The paper is very well written, and the messages are clear. I highlight below a few recommendations/questions for the authors, but want to emphasize my overall favorable impression of this work.

The authors note a 323-fold larger attack rate (in the abstract and the discussion), by comparing results from their survey in Yaounde to the case count data from Cameroon. Wouldn't a more fair comparison be their survey results to case count data from Yaounde?

In the methods, lines 248-250, the authors comment that the needed sample size of 250 was inflated to 1000 to increase precision. The statement is slightly misleading in that its not just four times the sample size, but rather increase of sample size while introducing sampling in clusters. I am not concerned because the target was a number of households equal to the original sample size, but a more accurate statement is that the inflation was largely to control for increased variance due to sampling in clusters. Alternatively, you can specify assumptions around the effect of clustering that will allow for statements on whether indeed this does increase precision.

On a similar theme, did you account for household clustering in the precision estimates? I don't have full access to the Lang-Reiczigel paper, but the snippets that I can read do not account for clustering. The prevalence results should account for clustering by household. This was done and is clearly stated for the logistic regression analyses.

Line 95 – I have never seen an IQR presented as symmetric; I suppose it is possible, but unlikely. Would encourage the authors to double check and to use a different format (like an interval) to report this result.

I personally find the presentation of results numerically and as figures in Figure 2 to be busy and duplicative. Aren't the numeric values sufficient?

Reviewer #1 (Remarks to the Author):

Nwosu et al. present results of a serological survey in one health province of urban Cameroon. The paper is well written and considered; I agree with the points made in the discussion. It is important research as relatively few serological surveys have been published from the continent of Africa, and none from Cameroon.

We thank the reviewer for the positive comments and the very helpful review of our work.

The statistical analyses are appropriate, though there are limitations that could be noted.

1) First, since sampling was within households, the survey population are not all independent. This is taken into account in the risk factor analysis, but not in the overall seroprevalence estimates, and could be noted as a potential source of bias.

We are grateful to the reviewer for catching this very important oversight. We have now updated the confidence-interval derivation protocol to take into account the sampling design. This is explained in the “Data Analysis” section of the Methods as follows:

“Confidence intervals around the crude- and age-sex-reweighted estimates are Wald-type intervals computed on the log-odds scale, as implemented in the R “survey” package;³⁴ these intervals take the survey’s single-stage cluster-sampled design into account. Confidence intervals for the test-adjusted estimates were calculated by bootstrap sampling using the “adjPrevSensSpecCI” function of the “bootComb” R package,³⁵ taking 100,000 parametric bootstrap samples for each estimate. The bootstrapping procedure propagates the uncertainty from the sensitivity and specificity validation studies, as well as the cluster-robust crude seroprevalence uncertainty into the final confidence interval for the adjusted seroprevalence. ”

We have also included a figure (Supplementary Figure 4) illustrating the uncertainty of the specificity, sensitivity and crude seroprevalence estimates, as well as the distribution of the bootstrapped estimates.

Supplementary Figure 4: A: Estimated densities of bootstrap parameters—prevalence, sensitivity and specificity—from their 95% confidence intervals B: Histogram of the sensitivity- & specificity-adjusted seroprevalence estimates from 100,000 bootstrapped samples.

2) Uncertainty in the sensitivity and specificity are not taken into account. On page 15, lines 272-273, I would like to see confidence intervals given for the sensitivity and specificity used.

Confidence intervals for the sensitivity and specificity have now been added in the second paragraph of the “Testing Procedure” section. These are Clopper-Pearson exact intervals.

These uncertainties are taken into account by the bootstrap method used for estimating the seroprevalence confidence intervals.

3) The hospitalised patients sensitivity is likely to be an over-estimate, firstly because patients will have experienced more severe hospitalised infection - giving rise to greater antibody responses on average, and secondly because all measurements were taken within 2 months - an optimal window during which we would expect no or little waning. The potential effect on adjusted estimates should at least be discussed. (Note again, I do believe the methods used are appropriate. To amend methods to address the above would be considerably more complex, and would not gain any further understanding.)

We thank the reviewer for bringing up this important caveat. This point is now noted as an important source of bias in the penultimate paragraph of the Discussion:

“A further limitation concerns the sensitivity value assumed for the IgG antibody test. This value was obtained from a validation study conducted with hospitalized COVID-19 patients, who are likely to have experienced more severe illness and greater antibody responses than our study population.²⁷ In addition, that study was conducted on samples obtained within two months of symptom onset, a short time frame over which minimal antibody waning would be expected. Our study, in contrast, was conducted about four months after the first epidemic peak in Cameroon, at which point the antibody levels of those infected may have dropped.³⁰ These two dissimilarities suggest that the assumed sensitivity may be an overestimate in the context of our study population. If that is the case, our final adjusted estimate may be downwardly-biased.”

On a related note, we have included a figure (Supplementary Figure 3) which shows the sampling time frame in the context of the case count epicurve. This should give the reader a clear sense of the lag between the first infection wave and the seroprevalence study.

Supplementary Figure 2. Nationally-reported COVID-19 cases counts per week and study sampling period.

The only other minor comment I have was on page 3, lines 70-72. I saw that results of a serological survey on a random sample of the Zambian population was published very recently in The Lancet Global Health (Mulenga et al). Please re-check the literature and update this sentence accordingly.

We have updated the sentence, and have added citation to this paper.

The final paragraph of the introduction now reads:

“However, only a few SARS-CoV-2 antibody serosurveys have been carried out in African countries to date,^{8,9,10,11,12,13,14} and the majority of serosurveys have been conducted on healthcare workers, convenience samples of blood donors and other non-representative populations....”

The 14th reference is to the Mulenga et al paper from Zambia.

Reviewer #2 (Remarks to the Author):

The manuscript reports the results of a cross-sectional sero-survey in Yaoundé, Cameroon. The authors adjust seroprevalence for test sensitivity and specificity and investigate the risk factors associated with IgG presence. This is one of the few Covid-19 sero-surveys from Africa.

The manuscript is lovely. Very clearly and well written, good exploration of risk factors and results are well discussed. It is great to see such detailed study from Africa, with some interesting results, e.g. very high overall seroprevalence compared to other parts of the world, especially at the time of the study.

We thank the reviewer for their very positive description of our work, and for their willingness to assist in its improvement.

My only main concern is the use of both univariate and multivariate analysis on the same dataset. I do not understand why this is done. Implementing the multivariate analysis from the start to all variables available would have been better and avoid multiple testing errors and removing important variables. Unless there was an unexplained reason for this, I would suggest removing the univariate analysis from the manuscript and present only the multivariate with all variables. If required, a backward selection method could be implemented to reduce the number of variables to the most essential ones.

We are grateful to the reviewer for pointing out this important issue. We agree that it is somewhat redundant to include both univariate and multivariate odds ratios. While arguably unnecessary however, presenting unadjusted odds has tended to be the norm in epidemiological studies of this kind. On this note, the journal editor has written the following to us:

“Please note that, although Reviewer 2 has requested that univariate results are not included in the paper, we appreciate that what you have done is standard practice for this sort of analysis (for example the STROBE statement says that both unadjusted and adjusted effects should be presented) and would ask that you keep both uni- and multivariate results.”

We share the reviewer’s concern about multiple testing errors with multiple univariate models. There is considerable debate about whether corrections (e.g. Bonferroni) are applicable here.¹ We ultimately decided not to perform any such corrections, but we have now noted this clearly in the fourth paragraph of the data analysis section. “...p-values were not corrected for multiple-hypothesis testing...”

1. Andrew Gelman, Jennifer Hill & Masanao Yajima (2012) Why We (Usually) Don't Have to Worry About Multiple Comparisons, Journal of Research on Educational Effectiveness, 5:2, 189-211, DOI: 10.1080/19345747.2011.618213

Other suggestions:

- It is not always clear when the authors are talking about raw vs adjusted seroprevalence, especially in the results. I would suggest keeping the raw values in the table as is but focus the text on the adjusted values only. Please make this clear throughout the text.

We have now addressed this.

Now, only the second and third paragraphs of the results section discuss the crude, unadjusted seroprevalence. And this section for the unadjusted estimates is labelled with a subheading.

The rest of the Results section focuses, as suggested, on the adjusted seroprevalence estimates.

- Also not clear what the age-sex weighted seroprevalence is, how it is estimated and how it is used. The methods on this are also not clear.

We thank the reviewer for pointing out the lack of clarity here. We have now updated the Methods section with the following paragraph describing the weighting procedure:

“To arrive at final seroprevalence estimates, crude proportions were first re-weighted to match the age-sex distribution of the Yaounde population, as sourced from the 2018 Cameroon DHS.³¹ Briefly, the sample was grouped into ten strata based on age (categorized as 5–14, 15–29, 30–44, 45–64 or 65+ years) and sex (categorized as male or female). Then weights for each age-sex stratum (e.g. males aged 5 to 14), were obtained by dividing the stratum’s actual population proportion by the stratum’s proportion in our sample (see Supplementary Table 1). As such, undersampled groups were given larger weights in the prevalence. Thus, the age-sex-standardized prevalences represent the expected prevalence if the age and sex distribution of our sample mirrored that of the reference population.”

We have added a table that shows the stratum-specific weights (Supplementary Table 1).

Supplementary Table 1: Sample sizes, strata and weights for each age-sex group. DHS stratum sizes refer to the estimated population from the 2018 DHS survey of Yaounde.

Age group	Sex	DHS stratum size	Sample stratum size	Stratum weight	Weight per individual
5 - 14	Female	50155.56	124	404	1.05
5 - 14	Male	48833.29	117	417	1.08
15 - 29	Female	67525.46	187	361	0.937
15 - 29	Male	62723.64	138	455	1.18
30 - 44	Female	44727.47	131	341	0.886
30 - 44	Male	46011.81	81	568	1.47
45 - 64	Female	21278.12	83	256	0.665
45 - 64	Male	21920.72	70	313	0.812
65 +	Female	5645.22	24	235	0.61
65 +	Male	5425.92	16	339	0.88

- Is the logistic regression implemented on the adjusted seroprevalence?

The regression is implemented on the crude IgG estimates, not the adjusted estimates. We have now noted this clearly in the new title for Figure 2: "Risk factor analysis for crude IgG seropositivity"

We feel that performing a regression on the crude, rather than the adjusted, estimates is appropriate because the sensitivity and specificity adjustments should not affect the reported odds ratios. That is, because the sensitivity and specificity of the test are (considered to be) the same for all groups, the adjustments for imperfect sensitivity and specificity will not change the relative odds of seropositivity for any group compared to any other. In addition, we are not aware of well-developed statistical methods to perform mixed-effects logistic regression on such adjusted estimates.

Including the sample weights in the regression is possible (and may shift certain odds ratios a little) but this is typically not done, and the methods for this are also not well developed for cluster-stratified samples.

Reviewer #3 (Remarks to the Author):

Thank you for the opportunity to review the paper titled “SARS-CoV-2 antibody seroprevalence and associated risk factors in an urban district of Cameroon.” This work is incredibly important, as the authors note that seroprevalence studies from Africa are rare. The paper is very well written, and the messages are clear. I highlight below a few recommendations/questions for the authors, but want to emphasize my overall favorable impression of this work.

We are heartened by the reviewer’s favourable comments and are thankful for the perceptive recommendations.

The authors note a 323-fold larger attack rate (in the abstract and the discussion), by comparing results from their survey in Yaounde to the case count data from Cameroon. Wouldn’t a more fair comparison be their survey results to case count data from Yaounde?

We agree that a comparison to the case count data from Yaoundé would be more appropriate. However, despite much searching, we have been unable to locate city-disaggregated case counts for Cameroon.

We believe the comparison to the national case count is still informative though. While our survey suggested that about **126,000** residents of Cité Verte (population **432,000**) had been infected with SARS-CoV-2, **only 24,000** cases had been reported in the whole country (population **26 million**) by the survey end date. This is certainly indicative of insufficient testing. We feel this discrepancy is important to highlight since it has important policy implications.

However, we have now adjusted the wording of the relevant Discussion paragraph to avoid implying that the 322-fold** difference is wholly explained by undertesting.

The paragraph now reads as follows:

“In this urban setting of Cameroon, the adjusted seroprevalence of SARS-CoV-2 IgG antibodies was found to be 29.2%, implying that around 126 000 of the district’s 432 858 inhabitants had been infected with SARS-CoV-2 by the survey’s end date, November 26, 2020. It is notable that by this date only 24,002 cases had been officially reported in the entire country, which has a population of 26.55 million people.¹⁵ Thus, the measured seroprevalence in the surveyed district was about 322 times greater than the nationwide attack rate implied by PCR- and antigen-confirmed case counts at the time.³ While this large discrepancy may indicate a particularly high attack rate in Yaoundé, it also likely points to inadequate testing, and suggests that the true cumulative incidence of COVID-19 in Cameroon may be much higher than the number of cases officially reported.”

** We changed the source for our population estimate for Cameroon, so the ratio is now 322, not 323.

In the methods, lines 248-250, the authors comment that the needed sample size of 250 was inflated to 1000 to increase precision. The statement is slightly misleading in that its not just four times the sample size, but rather an increase of sample size while introducing sampling in clusters. I am not concerned because the target was a number of households equal to the original sample size, but a more accurate statement is that the inflation was largely to control for increased variance due to sampling in clusters. Alternatively, you can specify assumptions around the effect of clustering that will allow for statements on whether indeed this does increase precision.

We are very grateful to the reviewer for catching this mistaken sentence. We have removed it and included, instead, a description of the single-stage cluster sampling design:

“We used a single-stage cluster sampling design with a target population of 250 households, and planned to interview all residents of each household—an estimated 1000 individuals.”

On a similar theme, did you account for household clustering in the precision estimates? I don't have full access to the Lang-Reiczigel paper, but the snippets that I can read do not account for clustering. The prevalence results should account for clustering by household. This was done and is clearly stated for the logistic regression analyses.

This is an important point which was also raised by reviewer 1. We copy our first response here:

We have now updated the confidence-interval derivation protocol to take into account the sampling design. This is explained in the “Data Analysis” section of the Methods as follows:

*“Confidence intervals around the crude- and age-sex-reweighted estimates are Wald-type intervals computed on the log-odds scale, as implemented in the *r* “survey” package;³⁴ these intervals take the survey's single-stage cluster-sampled design into account. Confidence intervals for the test-adjusted estimates were calculated by bootstrap sampling using the “adjPrevSensSpecCI” function of the “bootComb” R package,³⁵ taking 100,000 parametric bootstrap samples for each estimate. The bootstrapping procedure propagates the uncertainty from the sensitivity and specificity validation studies, as well as the cluster-robust crude seroprevalence uncertainty into the final confidence interval for the adjusted seroprevalence.”*

We have also included a figure (Supplementary Figure 4) illustrating the uncertainty of the specificity, sensitivity and crude seroprevalence estimates, as well as the distribution of the bootstrapped estimates.

Supplementary Figure 4: A: Estimated densities of bootstrap parameters—prevalence, sensitivity and specificity—from their 95% confidence intervals B: Histogram of the sensitivity- & specificity-adjusted seroprevalence estimates from 100,000 bootstrapped samples.

Line 95 – I have never seen an IQR presented as symmetric; I suppose it is possible, but unlikely. Would encourage the authors to double check and to use a different format (like an interval) to report this result.

This has been corrected.

I personally find the presentation of results numerically and as figures in Figure 2 to be busy and duplicative. Aren't the numeric values sufficient?

We agree that the figure is quite duplicative.

We do feel that the redundancy might be a small price to pay for the benefit of improved “glanceability”—the ease with which the main message can be gleaned.

Nonetheless, we have trimmed the figure slightly by removing the dot-whisker plots for the univariate results, leaving only the multivariate plot. Since the multivariate odds ratios are the most important estimates presented in this figure, we think it is helpful to highlight them graphically.

(We also noticed an error in our code which caused the inclusion of the neighbourhood variable in the regression analysis, even though this variable did not meet our threshold for inclusion. We have removed this from the regression figure. This had the happy side-effect of making the figure a little less busy.)

REVIEWERS' COMMENTS

Reviewer #1 (Remarks to the Author):

I would like to thank the authors for addressing my comments. The data analysis is adequate and the interpretation well reasoned. Given the limited serology data available from the African continent, this represents is an important piece of work.